# Antimicrobial Agent against Methicillin-Resistant *Staphylococcus aureus* Biofilm Monitored Using Raman Spectroscopy

**DOI:** 10.3390/pharmaceutics15071937

**Published:** 2023-07-12

**Authors:** Jina Kim, Young-Won Chin

**Affiliations:** College of Pharmacy and Research Institute of Pharmaceutical Sciences, Seoul National University, Seoul 08826, Republic of Korea; jakim0129@snu.ac.kr

**Keywords:** Methicillin-resistant *Staphylococcus aureus*, biofilm, antimicrobial agent, eugenol, Raman spectroscopy

## Abstract

The prevalence of antimicrobial-resistant bacteria has become a major challenge worldwide. Methicillin-resistant Staphylococcus aureus (MRSA)—a leading cause of infections—forms biofilms on polymeric medical devices and implants, increasing their resistance to antibiotics. Antibiotic administration before biofilm formation is crucial. Raman spectroscopy was used to assess MRSA biofilm development on solid culture media from 0 to 48 h. Biofilm formation was monitored by measuring DNA/RNA-associated Raman peaks and protein/lipid-associated peaks. The search for an antimicrobial agent against MRSA biofilm revealed that Eugenol was a promising candidate as it showed significant potential for breaking down biofilm. Eugenol was applied at different times to test the optimal time for inhibiting MRSA biofilms, and the Raman spectrum showed that the first 5 h of biofilm formation was the most antibiotic-sensitive time. This study investigated the performance of Raman spectroscopy coupled with principal component analysis (PCA) to identify planktonic bacteria from biofilm conglomerates. Raman analysis, microscopic observation, and quantification of the biofilm growth curve indicated early adhesion from 5 to 10 h of the incubation time. Therefore, Raman spectroscopy can help in monitoring biofilm formation on a solid culture medium and performing rapid antibiofilm assessments with new antibiotics during the early stages of the procedure.

## 1. Introduction

Infections caused by *Staphylococcus aureus* have emerged as one of the most significant global health problems. Various refractory infections are caused by *S. aureus*, primarily due to its ability to develop resistance to many drugs and form biofilm. Methicillin-resistant *S. aureus* (MRSA) is among the most dreadful clinical diseases. Every year, 20–50 new cases are diagnosed for every 100,000 people, resulting in a mortality rate of 40% [1]. Antibiotic-resistant bacteria, as well as their ability to build biofilms, pose a serious challenge to current medicine [2]. When the right antibiotic is identified in an effective time frame, thousands of lives can be saved, and drug-resistant bacteria will be slowed down [3]. Hence, the search for antibiotic treatment for drug-resistant bacteria and the exploration of MRSA biofilm is crucial.

Biofilms are forms of microorganisms attached to extracellular polymeric substances (EPS) that provide a home to bacterial cells [4]. Multi-resistant strains develop into biofilms due to harsh environments, such as a lack of oxygen and insufficient nutrients. In this extreme condition, EPS gives stability and a niche to microorganisms. It is mainly composed of polysaccharides, proteins, lipids, and extracellular DNA [5]. Biofilm formation involves complex developmental stages and usually follows three major phases: attachment, biofilm proliferation or maturation, and biofilm detachment or dispersal [6,7]. Biofilms emerge when planktonic cells permanently adhere to a surface and embed in an EPS. The sessile microbe proliferates and uses quorum sensing to grow into a stronger biofilm as the adhering bacterial cells are protected by the EPS [5]. Due to intense multicellular conglomerates, biofilms require a dose of antibiotics of up to 1000 times higher than that required for planktonic bacteria [8].

As biofilm infections are difficult to treat, rapid and easy diagnostic techniques are required to identify biofilm-forming multi-resistant *S. aureus* [9]. Traditionally, biofilms are identified by applying crystal violet dye, and their optical density (OD) is measured to calculate the mass [5,6]. The structure of biofilms can be studied using various techniques, such as scanning electron microscopy, transmission electron microscopy, atomic force microscopy, and confocal laser scanning microscopy [8]. Nevertheless, every method requires an in vitro cultivation of biofilms, which is time-consuming, costly, and labor-intensive [10]. Moreover, there is limited information on the interactions between strains and compounds in biofilms. Therefore, these methods cannot be applied in clinical diagnostics. Unlike other spectroscopies, Raman spectroscopy is label-free, quick, and contact-free. In this way, biomarker spectra can be obtained without invasive procedures. Bacteria can be identified using spectral information and statistical methods [11].

Raman spectroscopy was used in several bacterial studies, as microscopic biochemical differences can be accurately characterized, discriminated, and identified at species and subspecies levels for bacteria, fungi, and yeasts [3]. As all biologically associated molecules, such as proteins, nucleic acids, carbohydrates, and lipids, exhibit unique spectral features, Raman spectroscopy is a powerful technique for identifying different organisms [12,13]. The Raman spectrum contains several peaks corresponding to bacteria, including DNA, amino acids, carbohydrates, and lipids [14]. A chemometric approach was used to identify the correlation between Raman peaks and MRSA biofilm growth. To obtain a simple biological interpretation of the results, principal component analysis (PCA) was used and is a powerful multivariate analysis tool for exploring and interpreting high dimensional data in many fields [15,16,17,18]. The algorithm was used to determine if minor differences among MRSA biofilm cell structures could be differentiated based on their individual spectra. An analysis of loading plots can efficiently be performed to understand the relative contributions of biofilm formation [14,19]. The optimal multi-resistant biofilm formation time was determined based on the Raman spectrum.

Recently, as the advantages of Raman spectroscopy in microorganisms have been highlighted, the research on the direct measurement of bacteria inoculated on a solid medium has been increasing. Traditionally, liquid medium bacteria samples have weak Raman signals, which require costly and complex techniques such as SERS, and poor reproducibility. Alternatively, bacterial studies on solid media can be more traceable and durable, with less sample contamination and a stronger Raman spectrum. In 2022, Shen et al. presented a new fiber probe-based Raman technique on an agar plate to obtain more reliable and stable data for microorganism identification [20]. However, these studies focused only on planktonic bacteria and not biofilms. Here, the first direct observation of a biofilm on a solid medium was performed.

Raman spectral data collected from MRSA biofilms were used to test the antibacterial efficacy of natural products. Since MRSA biofilms are resistant to antibiotics, searching for treatment from medicinal plants is important. Eugenol, an antimicrobial compound with natural properties, could help reduce antibiotic use. Eugenol (4-allyl-2-methoxyphenol) has been used in various food products and cosmetics. In another study, eugenol was proven to be antimicrobial, antioxidant, anti-inflammatory, anticarminative, and antispasmodic [21,22,23,24,25]. Herein, eugenol exhibits antimicrobial activity against MRSA biofilms. The effect of eugenol on MRSA biofilms in different timelines was analyzed via Raman spectroscopy. The efficacy of eugenol was tested on different biofilm phases, and the most potent time to apply antibiotics to the biofilms was similar to that for colony-forming units (CFUs) on antibiofilm tests.

In this study, the growth of multi-resistant biofilm was monitored via in situ Raman spectroscopy. This study aimed to build a tool to directly monitor biofilms for hours, which allows the subsequent identification of the optimal time to eradicate growing biofilms with antimicrobial agents from a natural product. We have developed a new technique for the nondestructive monitoring of the growth of multi-resistant biofilms and to detect the optimum condition for effective antimicrobial activity. Therefore, the tool offers a simple, rapid, and inexpensive analysis of life-threatening clinical infections.

## 2. Materials and Methods

### 2.1. Materials

MRSA USA 300 was obtained from Professor Chung (Chung don won) Dongduk Women’s University in this study. All aqueous solutions were prepared using ultrapure water. Phosphate-buffered saline (PBS) and dimethyl sulfoxide were purchased from Thermo Fisher Scientific (Waltham, MA, USA). Tryptic soy broth (TSB; Difco, Becton Dickinson, and Company, Franklin Lakes, NJ, USA), tryptic soy agar (TSA; Difco, Becton Dickinson, and Company, Franklin Lakes, NJ, USA), and brain heart infusion agar (BHIA; Difco, Becton Dickinson, and Company, Franklin Lakes, NJ, USA) were used to culture bacteria. Congo red dye (Daejung chemicals, Siheung-si, Korea) and saccharose (Sigma Aldrich; Seoul, Korea) were used for a Congo red assay (CRA). Then, a 3M Petrifilm staph express count plate (3M, St. Paul, MN, USA), plastic spreader, and phosphate-buffered dilution water were all purchased from 3M to calculate CFUs per mL. The reagents and media were sterilized in an autoclave at 121 °C for 15 min before use. Vancomycin hydrochloride was purchased through Sigma Aldrich (Seoul, Korea). The antibiotic powders were dissolved in double-distilled water to produce antibiotic stock solutions (10 mg/mL). Eugenol was purchased through Tokyo Chemical Industry (Tokyo, Japan).

### 2.2. Preparation of Agar Plates

It is necessary to differentiate staphylococci based on their biofilm phenotypes to clarify their impact on infection diagnosis [26]. The phenotypic characterization of MRSA was carried on Congo red agar with 0.8 g of Congo red dye and 36 g of saccharose on 1.0 L of BHI agar [27]. First, an MRSA strain was grown in TSB at 36 °C for 24 h for activation. Then, 200 μL of sub-cultured microorganisms (OD600 = 1) was streaked on CRA and incubated at 37 °C for 24 h. The red color of the former colonies indicated negative results, while a black color indicated positive results [27].

In addition, a new agar plate was developed to serve as a substrate for growing biofilms to be analyzed via Raman spectroscopy. Modified brain heart infusion sucrose (MBHI-sucrose) agar was established for Raman spectroscopy sample preparation. Initially, biofilm on CRA plates was measured directly, but the fluorescence peaks from the staining obscured the peak, making accurate measurements impossible. Hence, various agar plates were compared to develop a Raman-optimized biofilm medium. To induce biofilm on an agar plate, Freeman et al. utilized a 5% sucrose addition to BHI broth [27]. As a result, the MBHI-sucrose agar plate included BHI broth, as well as 5% sucrose without Congo red dye. Next, 200 μL of MRSA was inoculated over MBHI-sucrose agar and incubated at 37 °C. Raman spectroscopy was used to detect biofilm growth on MBHI-sucrose agar from 0 to 48 h.

### 2.3. Preparation of Microorganism Sample

MRSA strains were individually grown in TSB at 37 °C for 24 h for activation. Then, 200 μL of sub-cultured microorganisms (OD 600 = 1) was inoculated on MBHI-sucrose agar. In order to monitor biofilm formation via Raman spectroscopy, the microorganisms were incubated on agar surfaces 0–48 h after inoculation. The incubation time refers to the duration the microorganisms were incubated on agar plates for prior to Raman measurements. Bacterial isolates were cultivated overnight on agar plates using the medium and incubation conditions described above, and were adjusted to an OD of 0.15 prior to incubation. After 6 h, samples were cultured and adjusted to OD values of 0.1, 0.3, and 1.0 to see whether or not the bacterial density (CFU/mL) altered the Raman spectra. At each OD, a five-fold serial dilution in PBS was conducted, and colonies were enumerated at a dilution that yielded between 10 and 30 colonies per 10 μL. To see whether or not the time of growth had an effect on the Raman spectra, samples were cultured hourly and adjusted to an OD of 0.3 to ensure a consistent bacterial density across samples. Following incubation, 1 mL was centrifuged at 9000 rcf for 3 min, the bacterial pellet was washed three times with PBS, and the supernatant was discarded. Dried drops were sonicated for 5 min. Each isolation was reproduced three times.

### 2.4. Antibiofilm Assay

To measure the efficacy of antibiotics against MRSA, minimum inhibitory concentrations (MICs) were determined. The MIC is the lowest concentration of antibiotics that inhibits the visible growth of microbes overnight [28]. The MICs of vancomycin and eugenol for MRSA were detected using the microdilution method. The minimum biofilm inhibition concentration (MBIC) is the lowest concentration of antibacterial agents that yields no colonies compared to the initial inoculum [29].

Eugenol was evaluated in an antibiofilm assay. MRSA biofilms pre-formed over 24 h were treated in a conical tube with 416 µg/mL (MIC) and 1024 µg/mL (2 × MIC) of eugenol. Biofilms prepared without eugenol were used as controls. To determine the CFU/mL of the biofilms, the biofilms were washed and counted [29]. All determinations were carried out in triplicates.

The effect of eugenol on biofilms was tested after 5 h and 36 h by rinsing the biofilms in PBS to remove planktonic bacteria and exposing them to varying eugenol concentrations at 37 °C. Then, the biofilms were again washed with PBS, sonicated, and dissolved in 200 μL of PBS. These diluted samples were plated and grown on MBHI-sucrose agar and viable CFUs per mL were determined. Petrifilm was used to check the biomass of the biofilms at different hours. Tests were conducted in triplicates, and the result was expressed as activity in terms of μg of extract per mL. In parallel to Raman experiments, the effects of antibiotics on cell viability were tested in liquid broths with the same inoculum and reagents. The experiments were triplicated on different dates.

### 2.5. Quantification of Biofilm Cell Counts on Petrifilm

Petrifilm count plates were used to evaluate the antibacterial effect of different concentrations of eugenol and vancomycin against MRSA. Antibiotic samples were ultrasonicated for 15 min before use. Each sample had an initial concentration of 1.0 OD at 600 nm prepared in a TSB medium. The MIC and MBIC of eugenol and vancomycin were added to the medium and incubated at 37 °C for 4 h at 180 rpm [30]. After incubation, Petrifilms were used to enumerate the efficacy of the antibiotics [31]. The samples were diluted 10 times, and 1 mL was inoculated on a 3M Petrifilm count plate and modified TSA and incubated at 37 °C for 24 h.

In accordance with 3M Petrifilm interpretation guidelines, bacterial colonies were counted after incubation. Red-violet colonies were counted as the total bacteria [31]. The inoculated sample on agar plates was analyzed via Raman spectroscopy to evaluate the antibacterial effect of antibiotics.

### 2.6. Raman Instrumentation

The experiments were conducted using Raman spectroscopy (RAMANtouch, Nanophoton Co., Osaka, Japan) with a single-mode diode laser at 785 nm. An objective microscope lens (Nikon, Tokyo, Japan; 50×; numerical aperture, 0.5; working distance, 500 μm) was used for focusing the laser beam. The laser spot dimension was approximately 0.6 μm × 1.0 μm. Raman-dispersed light was collected through the same objective lens, passed through a 50 μm pinhole slit, and scattered via holographic grating (300 lines/mm) onto a thermoelectrically cooled (−70 °C), deep-depleted, charge-coupled device (1340 × 400 pixels, Teledyne Princeton Instrument, Trenton, NJ, USA). Before and after experimental measurements, an internal silicon standard at 520 cm^−1^ was used to confirm system alignment and light throughput to the sample. A laser power of 30 mW and exposure time of 30 s were used for all Raman experiments.

### 2.7. Analyses and Characterizations

Analysis of variance was performed to determine the antimicrobial efficacies of different concentrations of vancomycin and eugenol in the solutions. All 171 spectra were collected for each experiment and the analysis was repeated three times. Procedures described by Association of Official Agricultural Chemists were used to calculate CFU/mL, which was converted into logarithms for statistical analysis [31].

The acquired Raman spectra were baseline-corrected using multiple scatter correction before performing the multivariate analysis of PCA using commercial unscrambler software 10.1 (Aspentech, Bedford, MA, USA). The PCA scores of the first and second principal components were used to plot 3D charts, based on which degree of similarity and difference of the Raman spectra of biofilm grown at different times was analyzed.

PCA algorithms were used to develop models that allow the discrimination and classification of planktonic bacterial cells into biofilm clusters using the spectral characteristics of MRSA biofilms. SIMCA-P 17.0 (Umetrics, Umeå, Sweden) software was used to conduct PCA and PLS-DA data analysis. Graphs were created using Origin 8.0 (OriginLab Corp., Northampton, MA, USA).

### 2.8. Data Preprocessing

Data preprocessing was conducted using Unscrambler software 10.1 (Aspentech, Bedford, NJ, USA) to baseline correct before the construction of chemometric models. The baseline correction helped to remove the background fluorescence from MRSA biofilm cells. In order to perform chemometrics, the Savitzky–Golay algorithm was performed to smooth the Raman spectra. Additionally, the standard normal variate (SNV) method was conducted to normalize the spectra.

## 3. Results and Discussion

### 3.1. Detection of the Biofilm Formation Phenotype by the CRA

The CRA developed by Freeman et al. is the standard method for observing biofilm formation. The biofilm phenotype is tested on Congo red agar, which identifies biofilm bacteria as black colonies and nonbiofilm bacteria as pink or red colonies [32]. MRSA was grown on a Congo red agar plate to confirm biofilm formation. In Appendix A, MRSA produced black colonies after 24 h of incubation. To observe the physical development of biofilms, the sample analysis was carried out on Congo red agar. MRSA inoculated for 1 h showed red pigmentation, but it developed biofilm on CRA agar after 24 h of incubation, and black pigmentation was observed.

### 3.2. Determination of MICs and MBICs to Test the Effectiveness of Antimicrobials against MRSA

The MIC is defined as the lowest dose of an antibiotic that inhibits the growth of bacteria following overnight incubation of the MRSA strain [33]. The MBIC is also known as the minimum biofilm inhibitory concentration [34]. Microdilution was used to detect the MICs and MBICs of vancomycin and eugenol. In this study, the control was untreated biofilms to represent the growth of bacterial cells. The MIC and twice the MIC of vancomycin were used as negative controls. Vancomycin remains one of the most effective drugs to treat MRSA infections with [35]. A natural compound, eugenol, was chosen as a potential conventional antibiotic capable of inhibiting the growth of planktonic and biofilm MRSA [25].

The MIC and MBIC values of eugenol were determined to test the efficiency of the best antibiotics for the bacterium. As a result, the MIC of vancomycin for MRSA was 2 µg/mL. Eugenol exhibited MIC values of of 416–1024 µg/mL against MRSA, and its MBIC values were twice that of the MIC [28,34].

### 3.3. Raman Spectral Signatures of MRSA Biofilms

Biofilm matrix components and their proportions are crucial to minimize and control biofilm formation. The matrix of biofilms typically consists of EPS, DNA, lipid, and extracellular vesicles [7,8]. Figure 1 shows the prominent features observed in the Raman spectra of MRSA biofilms that were phenotypically confirmed. Bacterial samples were observed directly on MBHI-sucrose agar via Raman spectra. The Raman spectra of growth media were analyzed to identify possible interference from the background. The Raman peak at 485 cm^−1^ in Figure 1 corresponded to monosaccharides, one of the ingredients in MBHI-sucrose agar, and did not overlap with the MRSA biofilm spectra [36].

Spectrum studies on specific chemical structures were selected from previously published studies (Table 1) [36,37,38,39,40,41,42,43,44,45]. Figure 1 illustrates the Raman spectrum of an average MRSA biofilm sample. The biofilm sample was at the beginning of the attachment on the agar plate, and Raman measurement was processed at 40 mW of laser power and a 50 s exposure time. This was repeated 10 times.

The characteristics of biofilm samples were less sharp spectral peaks than those of single-cell bacteria. Among the key peaks in biofilm samples were DNA/RNA-related peaks at 645 cm^−1^ assigned to the COO^−^ deformation of guanine, and at 785 cm^−1^ assigned to A and T (the ring in DNA/RNA bases).

The phenylalanine peak represents the existence of biofilms [40]. Peaks that are related to proteins appear as biofilms mature. The band at 1006 cm^−1^ represents the ring breathing of phenylalanine in a protein. The band at 856 cm^−1^ is attributed to the carbohydrate of C–C stretching [39]. The peaks at around 1273 and 1660 cm^−1^ are mainly attributed to amide I and amide III vibrations, and carboxylic acid stretching [37,42,43,44]. Most peaks were found in the bacterial Raman spectra of tryptophan at 1334 cm^−1^, which was used for protein biosynthesis. Moreover, the CH_3_CH_2_ twisting mode of lipids could be found at 1450 cm^−1^ [37,45].

MRSA strains can produce a golden carotenoid pigment called staphyloxanthin, which can serve as a potential antivirulence target [46]. The prominent two peaks at 1158 and 1526 cm^−1^ correlate to carotenoid bands, which represent C−C stretching and C=C stretching, respectively [40,45].

### 3.4. MRSA Biofilm Growth Monitored for 48 h via Raman Spectroscopy and a Microscope

A microscope was used to examine the biofilm cells. At a high magnification, single cells and a clump of cells adhered to the surface were detected (Figure 2a). The cells in the conglomerate generated yellowish slime, indicating the presence of EPS. The purpose of performing CRA was to validate the establishment of the biofilm (Appendix A). Microscopy images demonstrate that bacteria were primarily planktonic in the early stages of biofilm formation. The transformation of a three-dimensional biofilm occurs via reformation when the bacterial cells connect to the surface. Raman spectroscopy was applied to chemically characterize different multispecies biofilms using specific bands identified in the reference materials. Figure 2a shows representative microscopic images at different hours and the corresponding spectra of biofilms in Figure 2b. Based on the signature peaks of MRSA biofilms, the growth was monitored with Raman spectroscopy from 0 to 48 h. The dataset consists of 171 spectra of MRSA biofilms over a range of 250–1900 cm^−1^. Along with biofilm growth, microorganisms formed in different sizes and thicknesses. The Raman spectral data consist of different intensities and baselines due to chemical changes in biofilms over time. To correct this background spectral intensity at different hours, pre-processing was used to stabilize the baseline. The raw spectrum of biofilm growth was pre-processed using multiplicative scatter correction. Results showed that pre-processing can help in revealing the differences in the spectrum at different times.

The Raman spectrum of MRSA biofilms was acquired every 2 h for 48 h. In Figure 2b, each spectrum was an average of 10 random points at indicated hours. At 0 h, there was no significant peak observed from MRSA biofilms. The spectra obtained at 2 h and 4 h were similar to each other, and the peak at 485 cm^−1^ from monosaccharides in the culture medium was mainly observed in the spectrum. Starting from 4 h, very low-intensity carotenoid pigment peaks from MRSA strains started to appear at 1158 cm^−1^ and 1526 cm^−1^.

When the incubation time increased to 5 h, a different trend of the spectrum was observed. As the incubation time increased, the carotenoid and phenylalanine peaks increased. Especially at 6 h, peaks around 700–800 cm^−1^ appeared in the spectrum, which represents the DNA/RNA fragments from MRSA strains. DNA/RNA-related peaks increased throughout the initial incubation period. Since the bands at 645 cm^−1^ and 1525 cm^−1^ were observed in the guanine spectrum, their intensity may reflect the changes in the DNA or RNA concentrations of the biofilms [38].

From 5 h of incubation, the band at 1006 cm^−1^ from phenylalanine increased [47,48]. It was reported the phenylalanine bands could be used as the signature peak for biofilm formation [49]. Therefore, the increase in the 1006 cm^−1^ band suggests the formation of a MRSA biofilm structure. Based on these changes, it can be speculated that early biofilm maturation began after 5 h, and planktonic bacterial cells existed until 5 h of inoculation, which is the more disruptive state of bacterial colonies.

A microcolony was observed around 20 h, as indicated by the expression of carbohydrates and proteins. Biofilm matrix formation depends on fibrillary proteins present in the biofilm maturation process [49]. Microcolony formation, which was the attached bacteria beginning to replicate and encapsulate in EPS, was observed up until 24 h (Figure 2b). Carbohydrate bands (856 and 1158 cm^−1^) were found to be positively correlated with the in vitro formation of biofilms. It is also interesting to note that 24 h biofilms exhibit a relative reduction in nucleic acid band intensities compared with 5 h biofilms [47]. As illustrated in Figure 2b, the spectra from different structures of EPS matrix are characterized by strong polysaccharides [47,50,51] and lipopolysaccharides [51,52]. The initial formation of MRSA biofilms was in agreement with the CRA and microbiological analysis.

After 26–48 h of incubation, the overall intensity of the spectrum increased. Protein- and lipid-related bands at 1280, 1450, and 1660 cm^−1^ also increased significantly. However, the relative intensity of DNA/RNA fragment peaks decreased. The changes in the spectrum indicate that the metabolic activity of biofilms stopped, and the departure of bacteria started. Protein synthesis and lipid synthesis increase over time in response to environmental stress induced by nutrient depletion [53]. Bacteria respond to stress by increasing their protein and lipid synthesis. During the late stationary phase, bacterial cell metabolism ceases, and thus, no DNA, protein, or lipid synthesis occurs [54,55].

MRSA biofilm time points were cross-checked through viable counts (CFU/mL) and CRA. Therefore, the monitoring of bacteria through the Raman spectrum, which is correlated to biomarkers in biofilms, can be carried out to non-destructively analyze the entire biofilm life cycle. As the biofilm life cycle progresses, the intensity of bands changes, reflecting the changes in their concentration profiles. Biofilms are structures that surround bacterial cells and serve as a shell for living systems, limiting their interaction with the environment. This process continues until the completion of biofilms, and the process was in situ analyzed with Raman spectroscopy.

MRSA biofilm formation was confirmed using the Congo red agar assay and microbiological analysis. As shown in Figure 2b, MRSA biofilm formation started after approximately 5 h, which represented the beginning of the maturation of the biofilms. Although it was not possible to observe any changes in the growth curve after approximately 12 h using OD and viable counts, Raman spectroscopy was sufficiently sensitive to detect changes in bacteria from 0 h to 48 h. Unlike traditional measurements that only involve the concentration of cells in a sample, Raman spectroscopy can provide a “fingerprint” of the DNA, RNA, protein, and lipid content of the cells, which can be used to monitor the physiological changes in the samples [55].

### 3.5. PCA Plot to Classify Biofilm Growth Depending on Spectral Differences

To track the evolution of a biofilm community from a single cell to a multicellular level, an analytical tool was used to track multiple cells and clusters regarding their growth dynamics, size, and shape [56]. Biofilm growth was assessed via the spectral shift over time, as shown in Figure 2b. To confirm the prediction, PCA analysis was applied to the 171 spectral data. The Raman spectra of MRSA bacterial colonization and further growth of the colonizers on the surface, leading to structured microbial communities, could be more clearly identified and visualized by performing PCA on the Raman spectra, as shown in Figure 3A. PCA modeling was used to determine biomarkers to identify the indicators for assessing planktonic bacteria’s transition to biofilm maturation and categorization [57]. To acquire an initial overview of the dataset and to find patterns, a PCA analysis was performed. No samples fell outside of the 95% confidence ellipse, according to Hotelling T2 analysis. The findings of the analysis are supported by high levels of explained variation (R2X = 0.81) and predictive ability; Q2 (cum) = 0.97. In the PCA, approximately 99% of data variance is explained by the first two principal components: PC1, and PC2. Figure 3A shows the score plot of PC1, and PC2 of the growing MRSA biofilm, each of which accounts for 80.9%, and 16.7% of the variance, respectively. The data at different time points were demonstrated in three other groups to show the classification of biofilm states. 

Figure 3A shows the PCA of Raman spectra for MRSA biofilms with different cultivation times ranging from 0 to 48 h. The different groups represent the single-cell to multicellular level of biofilm and display significant changes in Raman peak intensities and peak ratios. MRSA biofilms at other metabolic states were clearly distinguished by the score plot. These classifications can be identified based on the following score plots: 0–5 h, which corresponds to the attachment of a single planktonic cell during biofilm formation (green); 6–24 h, which corresponds to biofilm proliferation during multicellular status (blue); and 26–48 h, which corresponds to the conglomerate of bacteria during biofilm dispersion (red). A separation of the groups was better between the second (6–24 h) and third time points (26–48 h) (Figure 3A). Bacterial cells in the first two groups overlapped slightly with biofilm colonies at the multicellular level. Poor group separation in the latter group of cell growth between the first and second groups indicated that there was an increasingly heterogeneous population of colonies from 5 to 10 h, with some colonies reversibly and irreversibly attaching to surfaces. However, there was a significant separation between the second and third groups of biofilm levels because in the proliferative state, biofilm cells actively underwent metabolic modifications, while metabolic cells remained inert after 24 h of biofilm development. Principal component (PC) loadings provided information on variables (wavenumber of the spectrum) that were important for group separation. By analyzing these plots, one can identify the most important variables in the dataset. The loading values of PCA are plotted in Figure 3B. Based on these plots, it is possible to distinguish microorganisms by their unique metabolic activities. The analysis of the loading plot leads to the differentiation of important factors attributed to biofilm biomarkers depending on time. Using the correlation coefficients (loadings) of component scores, spectral features important for planktonic bacteria to biofilm growth were identified and their contributions to each variable were determined using PC. Instead of examining a single wavenumber peak, a more chemically rich analysis is possible using regions of wavenumbers determined via correlation coefficients. It is more reliable to identify biomarkers important for discrimination when the selected spectral regions from loadings explain larger variances in data. As shown in Figure 3B, the PC2 loading plot shows no significant contribution from biofilm growth observed in the peak at 481 cm^−1^, which arises from monosaccharides in the medium. However, signature bands at 779, 849, 926, 1001, 1159, 1449, 1522, and 1652 cm^−1^ had high loading values. In particular, the hidden peak from 1449 (lipid) to 1652 cm^−1^ (amide I) was observed with a relatively high loading value via PCA. A significant intensity increase appeared for the amide I band of proteins, at 1600 cm^−1^, after biofilm growth for 26 h. The increases in protein and lipid peaks indicate the stress response to biofilm growth [58]. An increase in the protein concentration over longer incubation times can be explained by the connection between protein expression and biofilm formation [58]. Based on the PCA results, the Raman peaks demonstrate the most significant separation and absolute variances over time [54]. The loading peaks in PCA are highly correlated with the activation of DNA/RNA formation at 600–800 cm^−1^ [38]. After 5 h of biofilm growth, the signature peaks of MRSA carotenoids and the biofilm marker phenylalanine (at 1002 and 1159 cm^−1^, respectively) are prominent [40,41]. Lastly, the peaks that are different after 24 h of biofilm growth are correlated with lipids and amides at 1400–1600 cm^−1^, which are important EPS components of biofilms [37,45]. In Figure 3B, the most significant changes in spectral peaks during bacterial growth are presented. Raman analysis of these peaks can also be used to determine the metabolic state of unknown bacteria [53].

The results indicate that the PCA-based Raman technique can potentially identify and classify growing biofilm samples. The PCA method performed well in analyzing the Raman spectrum of the evolution of biofilm, and this analysis should be applicable in time-dependent antibiotic analysis. To resolve metabolic variations, a supervised analysis was performed in the overlapping 0–5 h and 6–24 h biofilm statuses. To categorize the MRSA biofilm cellular levels, the PLS-DA model was used (Appendix A). A receiver operating characteristic curve and permutation testing were used to verify the model (Appendix A).

### 3.6. Morphological Characterization of MRSA Biofilm Via Microscope and Biofilm Growth Assay Monitored in Time

In Figure 4a, the accumulated biomass of biofilm growth over time was calculated simultaneously with the *xy* mapping of the Raman spectra to characterize the morphological change. The microscopy image and the Raman map show representative bacterial accumulation on the BHIA agar surface at the time points (scale bars are 20 μm). Average MRSA biofilm Raman spectra corresponding to each time point are compared in Figure 4b.

Simultaneously with Raman monitoring, biofilm development from the MBHI-sucrose agar was measured hourly for 48 h to evaluate the early attachment of the bacteria, with three independent assessments carried out on different days. The biofilm growth curve computed the time-dependent buildup of biofilm biomass based on *xy* mapping images of the MRSA biofilm grown on a solid medium (Figure 4c). The MRSA biofilm growth curve showed a comparable growth angle to that of the typical bacterial growth curve. The slope grew slowly during the first 4 h, then exponentially from 5 to 24 h. The biofilm mass grew rapidly and displayed a high growth angle, perhaps owing to the attachment of bacteria to the surface as opposed to that of planktonic bacteria. Finally, the slope slowed down from 25 to 48 h. The three distinct growth rates in Figure 2 were closely associated with the PCA analysis. The *xy* mapping of Raman spectra was used to characterize the morphological change during biofilm growth. The Raman results showed spatial changes owing to the attachment of planktonic bacteria and the chemical alteration of biofilm at different time periods. Exemplary Raman images show a planktonic bacterial cell (1 h), initial biofilm cell attachment (4 h), biofilm colony maturation (12 h), and the conglomerate growth of biofilms on the agar surface (24 h). The morphological data and biofilm development curve show that in situ Raman spectroscopy can properly monitor biofilm progression.

### 3.7. Antimicrobial Activity of Eugenol against MRSA Biofilm Growth via Raman Spectroscopy and Biomass

In this research, the growth of MRSA biofilm was analyzed via PCA, and the beginning of the maturation process started after 5 h of incubation. The optimal time to eradicate biofilm was during the attachment stage of planktonic bacterial cells (0–5 h), and this was validated through the characterization of the eugenol–MRSA biofilm interaction using microbiological methods. Since eugenol is known to demonstrate strong antibiofilm properties against both MRSA and methicillin-sensitive Staphylococcus. aureus (MSSA) clinical strains, it was selected as a natural biofilm agent for rapid antibiofilm tests [26]. In Figure 5a,b, an antibiofilm assay was performed to determine the antibacterial activity of eugenol on MRSA within the biofilms. Biofilms exposed to 2 × MIC displayed strong bactericidal effects against MRSA [25]. After the treatment on biofilm grown for 5 h, counts of viable bacterial cells were decreased by more than 2-log_10_ and 4-log_10_. This observation suggests that the strong antimicrobial activity of eugenol on MRSA within biofilms reaches maximal effect toward 0–5 h of biofilm growth. The result is in agreement with the prediction.

Simultaneously with the antibiofilm assay, the effects on antibiotics on cell viability were tested on a MBHI-sucrose agar plate with the same inoculum and reagents used for the Raman experiments.

To figure out when spectral changes would occur, the time points were compared individually via Raman spectroscopy. Eugenol of the MIC was applied to monitor its killing effect on biofilms at different time points. In Figure 5a, MRSA biofilm grown for 5 h was treated with eugenol, and changes in the Raman spectrum were observed. The bands at 764, 1280, 1334, 1450, 1526, and 1650 cm^−1^ significantly decreased after the antibiotic treatment. The peaks were correlated to DNA/RNA fragments, tryptophan, lipid, carotenoid, and amide of MRSA biofilms. The Raman spectra of MRSA biofilm cultured within the first 5 h were observed with eugenol and vancomycin in different concentrations. When MRSA was treated with vancomycin of the MIC, the spectra decreased in intensity; however, the signature peaks did not decrease. Vancomycin was affected by the higher dosage, but it did not eradicate the biofilm. When treated with eugenol at the MBIC concentration, the peaks identified as 764, 1280, 1334, 1450, 1526, and 1650 cm^−1^ were significantly reduced in intensity and disappeared. The spectral change indicates that eugenol had an eradicating effect on the MBIC concentration. In addition, the results showed that with the decrease in bacterial concentration, the intensity of the Raman spectrum decreased. In Figure 5d, the linear correlation between the biomass of the biofilm and the Raman intensity is shown. Consequently, Raman spectroscopy could be used to measure antimicrobial activity against MRSA biofilms by observing the spectral changes in the Raman spectrum of bacterial cells.

A biofilm is a dominant form of microbial life, and it is difficult to eliminate completely. It provides protection to residing bacteria, so it is critical to find the best antibiotics in the most effective time frame. Therefore, the best time to apply eugenol was examined via Raman spectroscopy (Figure 5a). According to CFU calculations in Figure 5c, the number of biofilms counted was in agreement with the above results.

Eugenol demonstrated antibiofilm properties against clinical MRSA strains, especially in the early phase of biofilm formation. The biomasses of established biofilms were significantly decreased by the eugenol treatment. Similarly, the number of viable bacteria was significantly decreased in the eugenol-treated biofilms. Biofilm biomass was significantly decreased by 50% when eugenol was applied at the MIC [25]. A 4-log_10_ decrease in the number of viable cells was observed in biofilms treated with eugenol at twice the MIC. The MBIC of eugenol against MRSA biofilms was found to be twice the MIC value [30].

The antibacterial activities of eugenol were investigated against MRSA. Eugenol was evaluated for bactericidal activity by counting viable cells, providing a quantitative estimate of its efficacy. The inhibitory activities of eugenol and vancomycin are shown in Figure 5c. The samples were enumerated for bacterial viability at 24 h of exposure to antibiotics. The proliferation abilities of bacterial cells after treatment with the selected antibiotics at one and two times the MIC on biofilms at different hours are shown in Figure 5c. When MRSA biofilms grown for 36 h were exposed to eugenol at MIC, no significant difference was observed in the proliferation ability of MRSA. At two times the MIC, 7.3 logs CFU/mL of bacterial cells could proliferate, and at the MIC, 8.5 logs CFU/mL bacterial cells survived. However, the proliferation ability of MRSA biofilms decreased sharply when they were only grown for less than 5 h and treated with eugenol. After the antibiotic treatment on MRSA grown for 5 h, biofilms showed 4.3 logs CFU/mL cells at two times the MIC and 5.7 logs CFU/mL cells at the MIC. Similarly, 6.8 logs CFU/mL was observed for vancomycin treatments (Figure 5c). Based on these findings, the treatment with eugenol at an early stage of biofilm formation may enhance its bactericidal effects.

To evaluate the influence of antibiotics on the biofilm, the CFUs in MRSA biofilms were measured. The colony forming units/mL values in the presence of different antibiotics after different periods of biofilm growth are presented in the data. In the early stages, the antimicrobial activity of MIC eugenol was approximately 15% higher than that of vancomycin at the MIC. At double the MBIC value, eugenol caused the inhibition of more than 50% of the control before 5 h. However, the efficacy of eugenol on biofilms grown for 26 h was reduced only by 10% at the MBIC concentration.

Within an hour, our antibiofilm test was able to immediately detect the most efficient time to apply an antibiofilm agent on biofilm. This shows that eugenol is most potent during the attachment phase, as determined via Raman and CFU calculations. This recently identified natural antibiofilm compound is interesting prospects for innovative biofilm-associated infection treatments.

## 4. Prospects and Challenges

Raman spectroscopy can be used to analyze instantaneously the chemical properties of MRSA biofilms. It can compensate for traditional approaches such as CRA assays and crystal violet assays that permanently dye bacterial cells. Although the tool has advantages, the study still has limitations. The micro-scale analysis of Raman spectroscopy makes it difficult to capture the large range of complexity in biofilms. Since Raman spectroscopy is limited by the size of the sample, only a certain amount of material can be analyzed at a time. This means that only a small section of the biofilm can be studied. The Raman mapping is shown in Figure 4a to overcome this limitation. By combining the micro-scale analysis of Raman spectroscopy with Raman mapping, it was possible to capture the large range of complexity in biofilms. This technique is used to produce detailed chemical images of samples and can analyze be used to larger samples than can be analyzed via traditional Raman spectroscopy. Another challenge in the study was the limited number of bacteria samples. The study would have been more relevant to the broader range of microbes if more than one strain was used. Research using various strains can help determine which factors influence the microbial population and biofilm health. Additionally, comparing multiple strains gives a better understanding of how different microbes interact with each other and affect the biofilm. To ensure a more accurate representation of the microbial population, multiple bacterial strains should be studied in future research. Additionally, the study could have been improved by conducting more than one antibiotic agent efficacy assessment. This would have allowed for a better understanding of the effectiveness of the antibiotic agents. Additionally, the results would have been more reliable and comprehensive. We are planning to perform a screening of natural compounds on biofilms at different growth stages using Raman spectroscopy for further study. Despite the challenges to this research, Raman spectroscopy was advantageous to analyzing biological samples. For example, it allowed for a non-invasive and label-free approach to the analysis of biofilms and could provide information about the composition of the biofilm, changes in the biofilm over time, and the effect of drugs on the biofilm. Furthermore, Raman spectroscopy has the potential to provide insight into biological processes that would be difficult to access with traditional methods.

## 5. Conclusions

In this study, a novel approach was presented using Raman spectroscopy to identify new, safe, and effective agents with which to combat the increasing number of multi-resistant strains of bacteria. As MRSA forms a biofilm, the resistance against antibiotics increased and the natural compound eugenol was tested to inhibit the bacteria. There are several methods to evaluate bacteria, including crystal violet assays, Congo red agar assays, and SEM analysis. However, staining bacterial cells permanently can impact their functionality. In addition, it can be difficult to distinguish between different types of bacteria under the microscope, and results can be subjective. As a result, traditional staining techniques may not always be the best option for bacterial evaluation. Alternatively, Raman spectroscopy can be an effective option. It only takes a few minutes for the analysis to be completed, allowing the real-time monitoring of the results. This method allowed for a more accurate assessment of the biofilm structure as no staining or processing was required. Furthermore, it is a non-destructive technique, meaning that bacterial cells remain intact and functional. As a result, Raman spectroscopy can be used to predict initial MRSA biofilm adhesion to the surface and the optimal antibiotic administration time to avoid resistance caused by biofilm formation. MRSA biofilm formation was examined by measuring the changes in the Raman spectrum of biofilms and primary matrix materials, such as DNA, RNA, protein, lipid, and EPS. When analyzing the measured Raman spectra, PCA was used to predict the early stage of MRSA biofilm. The result showed that the planktonic MRSA bacteria attached to the surface formed a biofilm at approximately 5 h. To confirm the optimal eugenol treatment times, it was treated at 5 h, 24 h, and 36 h. The most effective MRSA biofilm growth could be inhibited by eugenol when treated before 5 h, and this result was consistent with the results based on CFU/mL. Based on this study, Raman spectroscopy could be used as an antimicrobial drug discovery tool to rapidly measure real-time biofilm formation without staining and suggest the optimal time for antibiotic administration to prevent biofilm formation.

## Figures and Tables

**Figure 1 pharmaceutics-15-01937-f001:**
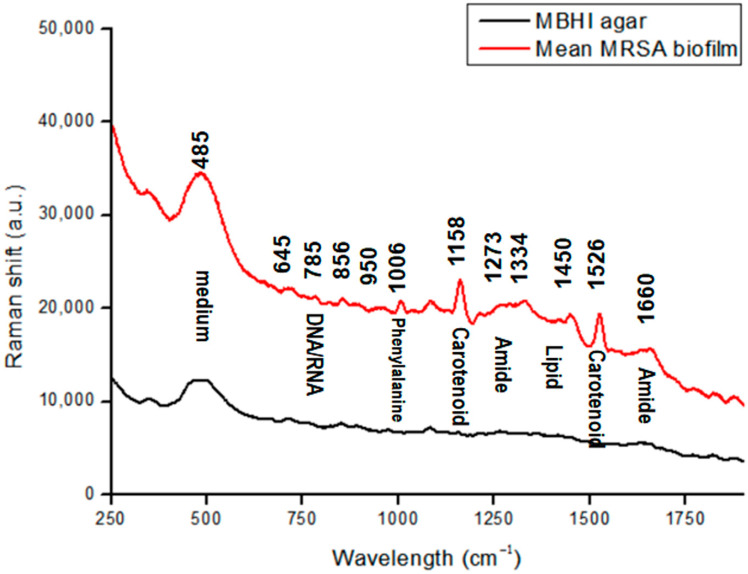
Raman spectra of MRSA biofilms in a spectral range of 250–1900 cm^−1^ (regions characteristic of chemical species are highlighted).

**Figure 2 pharmaceutics-15-01937-f002:**
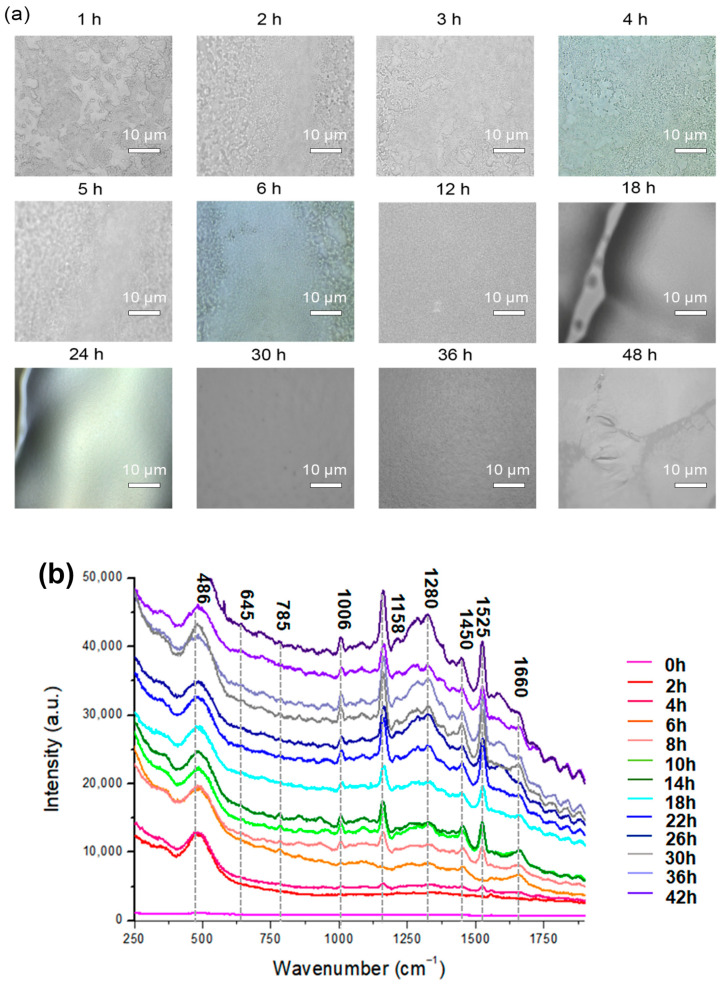
(**a**) Microscopic images of MRSA biofilms (0–48 h of biofilm). (**b**) Mean Raman spectra of MRSA biofilms every 2 h for 48 h. The region’s characteristic of chemical species is highlighted.

**Figure 3 pharmaceutics-15-01937-f003:**
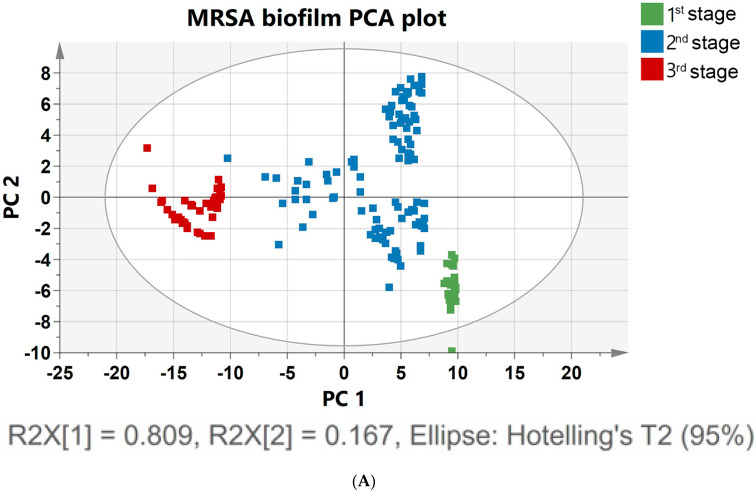
(**A**) PCA was constructed using the Raman spectra of MRSA biofilms with different cultivation times ranging from 0 to 48 h. The 171 samples of Raman spectra acquired at the planktonic bacteria (green), attachment of biofilm (blue), and multicellular levels of biofilm (red) are shown in PCA score plots. (**B**) Loading plot of MRSA biofilm (regions characteristic of chemical species are highlighted).

**Figure 4 pharmaceutics-15-01937-f004:**
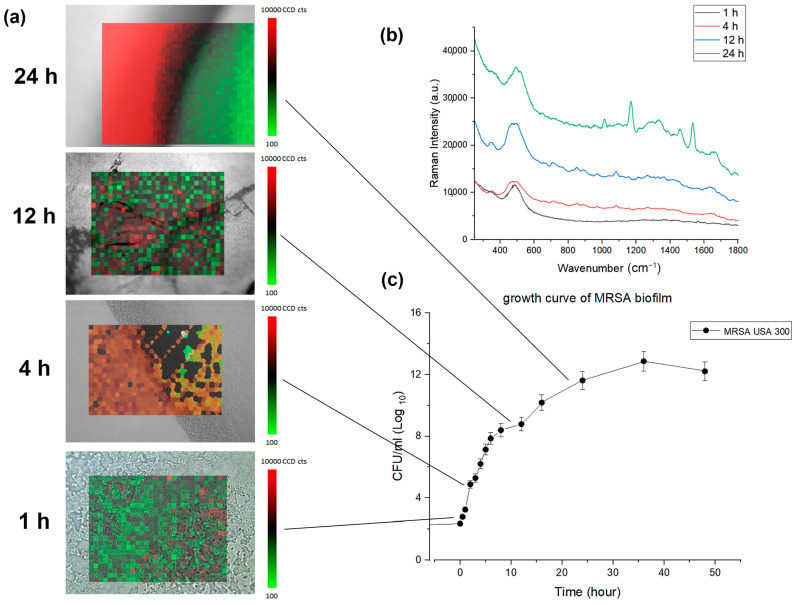
(**a**) Composite confocal Raman xy mapping images depicting the evolution of MRSA biofilm over time. 800 spectra (20 × 40, X × Y) were acquired across a 100 μm^2^ area averaging 3 times. (**b**) average MRSA biofilm Raman spectra corresponding to each time point (**c**) growth curve of MRSA biofilm formation.

**Figure 5 pharmaceutics-15-01937-f005:**
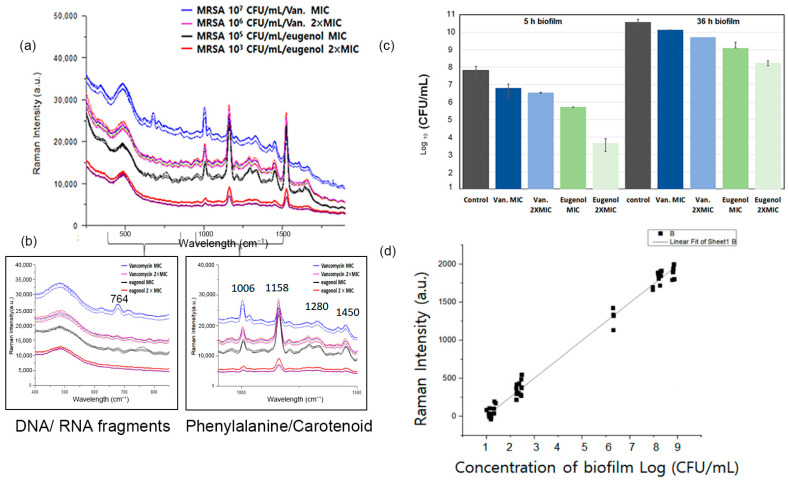
(**a**) Raman spectra of MRSA biofilms at different concentrations with eugenol and vancomycin treatment at MIC and MBIC. (**b**) Comparison of highlights of Raman spectral fragments. (**c**) Quantitative analysis of the antibacterial activity of the eugenol-loaded composites via direct contact for 5 h and 36 h against MRSA. (**d**) Linear correlation between logarithm of MRSA biofilm concentration (CFU/mL) and Raman intensity.

**Table 1 pharmaceutics-15-01937-t001:** Raman band assignments of MRSA biofilms.

NO.	Raman Feature (cm^−1^)	Assignments
1	485	Monosaccharide (medium)
2	645	COO^−^ deformation of guanine [36]
3	785	A (adenine ring of DNA/RNA base) [37]
4	856	C–C stretching from carbohydrates [38]
5	950	δ(C=C) [38]
6	1006	Phenylalanine ring [39]
7	1158	Carotenoid (C–C) [40]
8	1273	Amide III of proteins [41,42,43]
9	1334	Tryptophan [36]
10	1450	CH_2_ deformation of lipids [36,44]
11	1526	Carotenoid (C=C) [36]
12	1660	Amide I (α helix) [36]

## Data Availability

The data presented in the manuscript are available on request from the corresponding author.

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
