# Peer review of "Antimicrobial Agent against Methicillin-Resistant Staphylococcus aureus Biofilm Monitored Using Raman Spectroscopy"

_pharmaceutics, 2023, doi:10.3390/pharmaceutics15071937_

Round 1

Reviewer 1 Report

Overall this paper reports the use of raman spectroscopy to study MRSA biofilm production and demonstrates that raman can be used to observe alterations to the biofilm in the presence of eugenol. 

This work builds on an existing body of literature:

Eugenol is known to disrupt biofilm formation including that of MRSA

Raman has been used previously to study biofilm formation

PCA of raman biofilm spectra has been demonstrated before.

The novelty of the work is in combining these areas together. 

The authors have likely conducted the work in the manner described and the results are consistent with what i would expect the results to be. The work does however hinge on some PCA analysis which look a bit odd to me (see later on for further discussion of this). The authors should revisit the PCA to confirm that it's okay.

Overall, i suggest publication after editing. 

Line 209 "All data are presented as means of three replications." - how much spread is there in the replicate data? do the cluster in pca?

lines 221 and 222 are repetitive

"For MRSA strain" - be specific

" It is the most known lethal antibiotic treatment for MRSA. " - correct grammar and provide reference

"Natural compounds, such as eugenol, were chosen as potential conventional antibiotics capable of 

inhibiting the growth of planktonic and biofilm MRSA." Eugenol is already known to inhibit  biofilm growth which is presumably why you test it - reference a study which shows this

eg https://www.nature.com/articles/srep36377

https://doi.org/10.1371/journal.pone.0119564

You also mention 'compounds' but only discuss eugenol - which other compounds what did they show?

Font sizes in figures are generally too small for readability eg fig 3B

Figure 3a is strange in that it suggests taht some of the features are linearly correlated with the  and PC1 and PC2 are linearly correlated and not orthogonal. Linear trends in PCA can arise due to several reasons:

Correlated variables:  In PCA, the principal components are derived from linear combinations of the original variables. When variables are strongly correlated, their linear relationship is captured in the principal components, leading to linear trends in the PCA plot.

If there are systematic changes or trends present in the dataset, they can manifest as linear patterns in the PCA plot. For example, if the data represents a time series or a sequential process, the underlying trends or patterns can be reflected in the principal components, resulting in linear trends in the PCA analysis.

If the data represents different groups or classes, and there are consistent differences between these groups, PCA can capture these differences as linear trends. In such cases, the PCA plot may reveal distinct clusters or patterns corresponding to different groups, indicating separation along a linear trajectory.

The scaling or normalization applied to the data can affect the PCA results. If the variables are not properly normalized, it can introduce linear trends in the PCA plot. Normalization methods like centering and scaling should be applied appropriately to ensure that the PCA results accurately represent the underlying structure of the data. The authors should give details on how the normalisation is conducted.

It's important to note that linear trends in PCA do not necessarily indicate problems with the analysis. They can provide insights into the underlying structure of the data, especially if they align with known patterns or expected relationships. However, I'm not sure if the linear trends are expected in this work and therefore i would like the authors to discuss this in more detail - is there a systematic problem with the experiment? eg with your normalisation or with time points etc. what you may want to do (not necessarily for publication although maybe include in supplementary is to do the PCA of ALL collected data - ie don't take the mean of your replicates. Does this show anything? In your PCA you may also want to include some of the agar alone - these should cluster together and should be distinct from everything else in the experiment.  Not necessarily for this work but you may want to see what the biofilms look like with something like tSNE

The Results and discussion could benefit from better organization and clarity. Some sentences appear disjointed and could be rephrased for improved readability. 

The R&D mentions CFU calculations and the comparison of antibacterial activities between eugenol and vancomycin. However, it does not provide information about statistical analysis or significance testing. Including such details would strengthen the conclusions drawn from the data.

The R&D does not mention any limitations or potential sources of error in the study. It would be useful to discuss any limitations of the experimental approach or potential confounding factors that may affect the interpretation of the results. While the text does not explicitly mention the limitations of the study, there are several potential limitations that could be considered:

The study has used a limited number of strains (n =1  ) , which could affect the generalizability of the findings. The variability among different strains , as well as the heterogeneity within biofilm communities, could impact the applicability of the results to broader microbial populations.

Likewise, the laboratory conditions may not fully capture the complexity and diversity of natural biofilms. 

Limited assessment of antibiotic efficacy: The study focused on evaluating the effects of eugenol as an antibiofilm agent. However, the efficacy of other antibiotics or antimicrobial agents was not thoroughly investigated. The conclusions should be interpreted within the context of eugenol's specific effects and may not be generalizable to other antimicrobial agents.

The study focuses on relatively short-term biofilm growth and antibiotic treatments. Long-term effects, such as the potential for biofilm reformation or the development of resistance over extended periods, were not investigated. Considering longer observation periods might provide a more comprehensive understanding of biofilm dynamics and antibiotic resistance. 

The advantages and limitations of Raman spectroscopy compared to traditional biofilm assessment methods were not discussed. A comparison with conventional techniques would provide a more comprehensive evaluation of the applicability and utility of Raman spectroscopy in this context. This leads you to call the method 'rapid' but you dont report any time comparison of the experiments 

I am a native English speaker. The standard of English is good with only a couple of minor grammatical errors which the authors should be able to correct easily. 

Author Response

Comments from the reviewer 1:

Overall this paper reports the use of raman spectroscopy to study MRSA biofilm production and demonstrates that raman can be used to observe alterations to the biofilm in the presence of eugenol. 

This work builds on an existing body of literature:

Eugenol is known to disrupt biofilm formation including that of MRSA

Raman has been used previously to study biofilm formation

PCA of raman biofilm spectra has been demonstrated before.

The novelty of the work is in combining these areas together. 

The authors have likely conducted the work in the manner described and the results are consistent with what i would expect the results to be. The work does however hinge on some PCA analysis which look a bit odd to me (see later on for further discussion of this). The authors should revisit the PCA to confirm that it's okay.

Overall, i suggest publication after editing. 

Author response: We thank the reviewer for these valuable comments. It is correct that we have conducted this work based on some existing literature. In addition, it was the first study to analyze MRSA biofilm growth using Raman spectroscopy and to investigate the effectiveness of eugenol in disrupting biofilm through this technique. However, PCA analysis will be revisited to clarify the result.   

Line 209 "All data are presented as means of three replications." - how much spread is there in the replicate data? do the cluster in pca?

We agree that the sentence lacks accuracy. We have edited the sentence to indicate there were 171 spectra in the data and repeated it in three batches. (p.5, line 209, 210)

“All 171 spectra were collected for each experiment and it was repeated three times.”

lines 221 and 222 are repetitive

Thank you for the comment. We have deleted the sentence (p. 5, line 221)

"For MRSA strain" - be specific

We agreed to the comment and edited the sentence, (p. 5, line 235, 236)

“The MIC is defined as the lowest dose of antibiotic that inhibits the growth of bacteria following overnight incubation of the MRSA strain [33].”

" It is the most known lethal antibiotic treatment for MRSA. " - correct grammar and provide reference

We have edited the sentence and added a reference to the sentence. (p. 5, line 240, 241)

“Vancomycin remains one of the most effective drugs to treat MRSA infections [35].”

"Natural compounds, such as eugenol, were chosen as potential conventional antibiotics capable of inhibiting the growth of planktonic and biofilm MRSA." Eugenol is already known to inhibit biofilm growth which is presumably why you test it - reference a study which shows this

eg https://www.nature.com/articles/srep36377

https://doi.org/10.1371/journal.pone.0119564

Thank you for your considerate comment, I have added the reference in this sentence. (p. 6, 241-243)

“Natural compounds, such as eugenol, were chosen as potential conventional antibiotics capable of inhibiting the growth of planktonic and biofilm MRSA [25].”

You also mention 'compounds' but only discuss eugenol - which other compounds what did they show?

It is true I mention compounds in this sentence. However, eugenol and vancomycin were the only compounds compared in this research. (p. 6, 241-243)

“A natural compound, such as eugenol, was chosen as potential conventional antibiotics capable of inhibiting the growth of planktonic and biofilm MRSA [25].”

Font sizes in figures are generally too small for readability eg fig 3B

We agree with the comment and edited the font sizes of the figures.

Correlated variables:  In PCA, the principal components are derived from linear combinations of the original variables. When variables are strongly correlated, their linear relationship is captured in the principal components, leading to linear trends in the PCA plot.

 If there are systematic changes or trends present in the dataset, they can manifest as linear patterns in the PCA plot. For example, if the data represents a time series or a sequential process, the underlying trends or patterns can be reflected in the principal components, resulting in linear trends in the PCA analysis.

If the data represents different groups or classes, and there are consistent differences between these groups, PCA can capture these differences as linear trends. In such cases, the PCA plot may reveal distinct clusters or patterns corresponding to different groups, indicating separation along a linear trajectory.

The scaling or normalization applied to the data can affect the PCA results. If the variables are not properly normalized, it can introduce linear trends in the PCA plot. Normalization methods like centering and scaling should be applied appropriately to ensure that the PCA results accurately represent the underlying structure of the data. The authors should give details on how the normalisation is conducted.

It's important to note that linear trends in PCA do not necessarily indicate problems with the analysis. They can provide insights into the underlying structure of the data, especially if they align with known patterns or expected relationships. However, I'm not sure if the linear trends are expected in this work and therefore i would like the authors to discuss this in more detail - is there a systematic problem with the experiment? eg with your normalisation or with time points etc. what you may want to do (not necessarily for publication although maybe include in supplementary is to do the PCA of ALL collected data - ie don't take the mean of your replicates. Does this show anything? In your PCA you may also want to include some of the agar alone - these should cluster together and should be distinct from everything else in the experiment.  Not necessarily for this work but you may want to see what the biofilms look like with something like tSNE

We are grateful for your guidance and examples in reviewing PCA. PCA data might have presented a linear pattern since the experiment was time series-based. Even so, PCA revealed clearer clustering after normalizing and scaling the data. We were able to gain a better visual representation of the clusters with the correction method, making the PCA result more useful and insightful. For further study, we will definitely consider assessing with tSNE for future work. The manuscript was revised by adding a section on preprocessing data and showing the PCA results with the updated results. (p. 5, line 223-231; data pre-processing, p. 10, 11, line 390-397; Fig. 3(a) and (b))

2.8       Data preprocessing

Data preprocessing was conducted using unscrambler software (Aspentech, Bedford, USA) to baseline correct before the construction of chemometric models. The baseline correction helped to remove the background fluorescence from MRSA biofilm cells. In order to perform chemometrics, Savitzky–Golay algorithm was performed to smooth the Raman spectra. Additionally, Standard Normal Variate (SNV) method was conducted to normalize the spectra.

Figure 3. (A) PCA was constructed using the Raman spectra of MRSA biofilms with different cultivation times ranging from 0 to 48 h. The 171 samples of Raman spectra acquired at the planktonic bacteria (blue), attachment of biofilm (green), and multicellular level of biofilm (red) are shown in 3D PCA score plots. (B) Loading plot of MRSA biofilm (regions characteristic of chemical species are highlighted).

The Results and discussion could benefit from better organization and clarity. Some sentences appear disjointed and could be rephrased for improved readability. 

Thank you for giving advice to improve the manuscript.  We have observed the results and discussion section and have made some changes to improve readability.

The R&D mentions CFU calculations and the comparison of antibacterial activities between eugenol and vancomycin. However, it does not provide information about statistical analysis or significance testing. Including such details would strengthen the conclusions drawn from the data.

We agree that the CFU calculations of antibacterial activities were not highlighted enough. However, we do elaborate on the data in Figure 5C. We will attach the enlarged figure to show details of the CFU comparison.  

“The inhibitory activities of eugenol and vancomycin are shown in Figure 5C. The samples were enumerated for bacterial viability at 24 h of exposure to the antibiotics. The proliferation abilities of bacterial cells after treatment with the selected antibiotics at one and two times the MIC on biofilms at different hours are shown in Figure 5C. When MRSA biofilms grown for 36 h were exposed to eugenol at MIC, no significant difference was observed in the proliferation ability of MRSA. At two times the MIC, 7.3 logs CFU/mL bacterial cells could proliferate, and at MIC, 8.5 logs CFU/mL bacterial cells survived. However, the proliferation ability of MRSA biofilms decreased sharply when only grown for less than 5 h and treated with eugenol. After the antibiotic treatment on 5h grown MRSA, biofilms showed 4.3 logs CFU/mL cells at two times the MIC and 5.7 logs CFU/mL cells at MIC. Similarly, 6.8 logs CFU/mL were observed for vancomycin treatments (Figure 5C). Based on these findings, the treatment with eugenol at an early stage of biofilm formation may enhance its bactericidal effects.”

Figure 5(C) Quantitative analysis of the antibacterial activity of the eugenol-loaded composites by direct contact for 5 h and 36 h against MRSA

The R&D does not mention any limitations or potential sources of error in the study. It would be useful to discuss any limitations of the experimental approach or potential confounding factors that may affect the interpretation of the results. While the text does not explicitly mention the limitations of the study, there are several potential limitations that could be considered:

The study has used a limited number of strains (n =1  ) , which could affect the generalizability of the findings. The variability among different strains , as well as the heterogeneity within biofilm communities, could impact the applicability of the results to broader microbial populations.

Likewise, the laboratory conditions may not fully capture the complexity and diversity of natural biofilms. 

Limited assessment of antibiotic efficacy: The study focused on evaluating the effects of eugenol as an antibiofilm agent. However, the efficacy of other antibiotics or antimicrobial agents was not thoroughly investigated. The conclusions should be interpreted within the context of eugenol's specific effects and may not be generalizable to other antimicrobial agents.

The study focuses on relatively short-term biofilm growth and antibiotic treatments. Long-term effects, such as the potential for biofilm reformation or the development of resistance over extended periods, were not investigated. Considering longer observation periods might provide a more comprehensive understanding of biofilm dynamics and antibiotic resistance. 

The advantages and limitations of Raman spectroscopy compared to traditional biofilm assessment methods were not discussed. A comparison with conventional techniques would provide a more comprehensive evaluation of the applicability and utility of Raman spectroscopy in this context. This leads you to call the method 'rapid' but you dont report any time comparison of the experiments 

We thank you for pointing out that the R&D does not mention any limitations or potential sources of error in the study. We have accepted your considerate suggestions and discussed the limitations of this study and added the section on prospects and challenges. (p. 15, 563-593)

“4. Prospects and Challenges

Raman spectroscopy can analyze instantaneously the chemical properties of MRSA biofilms. It can compensate for traditional approaches such as CRA assay and crystal violet assay that permanently dye bacterial cells. Although the tool has advantages, the study still has limitations. The micro-scale analysis of Raman spectroscopy makes it difficult to capture the large range of complexity in biofilms. Since Raman spectroscopy is limited by the size of the sample, only a certain amount of material can be analyzed at a time. This means that only a small section of the biofilm can be studied. The Raman mapping is shown in Figure 4A to overcome this limitation. By combining the micro-scale analysis of Raman spectroscopy with Raman mapping, it is possible to capture the large range of complexity in biofilms. This technique is used to produce detailed chemical images of samples and can analyze larger samples than traditional Raman spectroscopy.

Another challenge in the study was a limited number of bacteria samples. The study would have been more relevant to the broader range of microbes if more than one strain was used. Research using various strains can help determine which factors influence the microbial population and biofilm health. Additionally, comparing multiple strains gives a better understanding of how different microbes interact with each other and affect the biofilm. To ensure a more accurate representation of the microbial population, multiple bacterial strains should be studied in future research. Also, the study could have been improved by conducting more than one antibiotic agent efficacy assessment. This would have allowed for a better understanding of the effectiveness of the antibiotic agents. Additionally, the results would have been more reliable and comprehensive. We are planning to perform screening of natural compounds on biofilms at different growth stages using Raman spectroscopy for further study. Despite of challenges to this research, Raman spectroscopy was advantageous to analyze biological samples. For example, it allowed for a non-invasive and label-free approach to the analysis of biofilms and could provide information about the composition of the biofilm, changes in the biofilm over time, and the effect of drugs on the biofilm. Furthermore, Raman spectroscopy has the potential to provide insight into biological processes that would be difficult to access with traditional methods.” 

Reviewer 2 Report

Methicillin-resistant Staphylococcus aureus (MRSA) is a leading cause of infections and forms biofilms on medical devices, making them resistant to antibiotics. Authors used Raman spectroscopy to study MRSA biofilm development on solid culture media over a 48-hour period. They monitored biofilm formation by measuring DNA/RNA and protein/lipid-associated Raman peaks. The study identified Eugenol as a promising antimicrobial agent for breaking down MRSA biofilms. The optimal time for inhibiting the biofilm was found to be within the first 5 hours of formation. The researchers also used Raman spectroscopy coupled with Principal Component Analysis (PCA) to distinguish planktonic bacteria from biofilm conglomerates. The results indicated early adhesion of the biofilm between 5 to 10 hours of incubation. The manuscript can be accepted for publication after minor revision.

1. Please describe if you did any postprocessing to Raman spectra - averaging, smoothing etc?

2. line 222-223 - both graphs and plots were made via Origin so it could be one sentence

3. Figure 1 - could Authors remove the background from the Raman spectra?

4. Figure 2 - please add scale to the images. The quality of the Raman spectra plots could be improved. The spectra could be normalized and the backgrouound could be removed.

5. Figure 4 and Figure 5 - please increase the quality of the figure.

6. In Conclusions Authors should elaborate more on their method - could they compare this method with other methods?

The english is good, however the Authors should re-read it again carefully.

Author Response

Reviewer 2.

Methicillin-resistant Staphylococcus aureus (MRSA) is a leading cause of infections and forms biofilms on medical devices, making them resistant to antibiotics. Authors used Raman spectroscopy to study MRSA biofilm development on solid culture media over a 48-hour period. They monitored biofilm formation by measuring DNA/RNA and protein/lipid-associated Raman peaks. The study identified Eugenol as a promising antimicrobial agent for breaking down MRSA biofilms. The optimal time for inhibiting the biofilm was found to be within the first 5 hours of formation. The researchers also used Raman spectroscopy coupled with Principal Component Analysis (PCA) to distinguish planktonic bacteria from biofilm conglomerates. The results indicated early adhesion of the biofilm between 5 to 10 hours of incubation. The manuscript can be accepted for publication after minor revision.

Thank you very much for acknowledging the importance of this study and giving us the opportunity to revise and submit it for publication.

  1. Please describe if you did any postprocessing to Raman spectra - averaging, smoothing etc?

Thank you for your comment. We have added the section to elaborate on the normalization of Raman spectra (p. 5, line 223-231; data pre-processing)

2.8       Data preprocessing

Data preprocessing was conducted using unscrambler software (Aspentech, Bedford, USA) to baseline correct before the construction of chemometric models. The baseline correction helped to remove the background fluorescence from MRSA biofilm cells. In order to perform chemometrics, Savitzky–Golay algorithm was performed to smooth the Raman spectra. Additionally, Standard Normal Variate (SNV) method was conducted to normalize the spectra.

  1. line 222-223 - both graphs and plots were made via Origin so it could be one sentence

We agree with the advice. We have deleted the sentence (p. 5, line 221)

  1. Figure 1 - could Authors remove the background from the Raman spectra?

Thank you for the point out. We have gone ahead and removed the background from Figure 1.

  1. Figure 2 - please add scale to the images. The quality of the Raman spectra plots could be improved. The spectra could be normalized and the backgrouound could be removed.

We have edited the images to follow the given advice. We have improved the quality of the plots in Figure 2(B). The background was removed and the spectra were normalized for the PCA evaluation. The spectra were left without normalization in this figure due to the overlapping of the peaks to highlight the changes over time.   

  1. Figure 4 and Figure 5 - please increase the quality of the figure.

We will increase the quality of the figures.

  1. In Conclusions Authors should elaborate more on their method - could they compare this method with other methods?

We thank the reviewer for the valuable suggestion. Our conclusion is now supported with more on the method that strengthens our arguments. (p. 16, line 595-601)

“There are several methods to evaluate bacteria, including crystal violet assay, Congo red agar assay, and SEM analysis. However, staining bacterial cells permanently can impact their functionality. In addition, it can be difficult to distinguish between different types of bacteria under the microscope, and results can be subjective. As a result, traditional staining techniques may not always be the best option for bacterial evaluation. Alternatively, Raman spectroscopy can be an effective option. It only takes a few minutes for the analysis to be completed, allowing real-time monitoring of the results. This method allowed for a more accurate assessment of the biofilm structure as no staining or processing was required. Furthermore, it is a non-destructive technique, meaning that bacterial cells remain intact and functional.”